# The Immune Response in the Uteri and Placentae of *Chlamydia abortus*-Infected Ewes and Its Association with Pregnancy Outcomes

**DOI:** 10.3390/pathogens12060846

**Published:** 2023-06-19

**Authors:** Sergio Gaston Caspe, David Andrew Ewing, Morag Livingstone, Clare Underwood, Elspeth Milne, Neil Donald Sargison, Sean Ranjan Wattegedera, David Longbottom

**Affiliations:** 1Moredun Research Institute, Penicuik EH26 0PZ, UK; 2Estación Experimental Mercedes, Instituto Nacional de Tecnología Agropecuaria (INTA), Corrientes W3400, Argentina; 3Biomathematics and Statistics Scotland, Edinburgh EH9 3FD, UK; 4Royal (Dick) School of Veterinary Studies, University of Edinburgh, Edinburgh EH10 5HF, UK

**Keywords:** enzootic abortion of ewes, ovine enzootic abortion, *Chlamydia abortus*, immune response, maternofoetal interface, uterus, placenta, immunohistochemistry, in situ hybridisation

## Abstract

The enzootic abortion of ewes, caused by the bacterium *Chlamydia abortus* (*C. abortus*), is one of the main causes of abortion in sheep. There are multiple contributory factors, including chlamydial growth, host immune response, and hormonal balance, that result in different pregnancy outcomes, such as abortion, the birth of weak lambs that may die, or healthy lambs. This study aimed to determine the relationship between phenotypical patterns of immune cell infiltration and different pregnancy outcomes in twin-bearing sheep (both lambs born dead; one alive and one dead; both alive) when experimentally infected with *C. abortus*. Both the sheep uteri and placentae were collected after parturition. All samples were analysed for specific immune cell features, including cell surface antigens and the T-regulatory (Treg) cell-associated transcription factor and cytokines, by immunohistochemistry and in situ hybridisation. Some of these immunological antigens were evaluated in ovine reproductive tissues for the first time. Differential patterns of T helper/Treg cells revealed significant group effects in the placentae. It suggests the potential role that the balance of lymphocyte subsets may play in affecting different pregnancy outcomes in *C. abortus*-infected sheep. The present study provides novel detailed information about the immune responses observed at the maternofoetal interface in sheep at the time of pre-term abortion or lambing.

## 1. Introduction

The enzootic abortion of ewes (EAE) is considered one of the most important infectious causes of abortion in ewes worldwide (except for Australia and New Zealand) [1]. One of the most noticeable characteristics of EAE is the different pregnancy outcomes that occur in different sheep, including pre-term abortion, the birth of weak lambs that can fail to survive beyond 48 hrs., the lambing of apparently “healthy” lambs, or a combination of these [2]. Sheep can be infected mainly through the oronasal route prior to or during pregnancy by the intracellular Gram-negative bacterium *Chlamydia abortus* (*C. abortus*). The progression of EAE appears to be synchronised with the mid-stage of gestation, whereby the chlamydiae gravitate and multiply in the chorionic epithelial lining of the foetal placental cotyledon, from which they can disseminate to the surrounding tissue of the intercotyledonary membranes [2]. However, although there is a general relationship between the degree of placental lesions and pregnancy outcome [3], higher gross placental lesions are not always associated with abortion events [4,5]. Such abortions result from a combination of events that are associated with the pathogenesis of EAE, including necrosis of the placenta resulting from bacterial multiplication, degradation, and the release of immunostimulatory chlamydial membrane components, such as lipopolysaccharide (LPS), changes in the hormonal balance, and maternal/foetal immune response [6].

For successful pregnancies across eutherian mammals, a combination of physiological, hormonal, and immunological features was integral to the delivery of live young [7,8,9]. The prominence of these features has been most widely described for humans and small laboratory rodents, while less is known about the features supporting pregnancy in ruminants, including sheep [6,9]. The physiology of the maternofoetal interface varies widely between species. It was hypothesised that its variability between species could affect the requirement of immunological features that support pregnancy [8,9]. In haemochorial placentation (humans, rats, and mice), the immune modulation in the maternofoetal interface is essential for maintaining the pregnancy post-implantation of the conceptus. However, in less-invasive placentation such as synepitheliochorial (sheep), the requirement for local immune regulation might be lower. Wattegedera et al. [10] found a notable difference in the major histocompatibility complex class I transcription, indoleamine 2,3-dioxygenase 1 expression, and placental natural killer (NK) cell distribution in sheep compared to species with haemochorial placentation.

In humans and mice, different subsets of a cluster of differentiation (CD) 4 T helper (Th) cells, such as Th1, Th2, Th9, and Th17, have a distinct roles in maintaining pregnancy [11]. While Th1 immunity is dominant in controlling the trophoblast invasion during the period of implantation, the immune response shifts to Th2 immunity after placental implantation [11,12,13]. The dominance of a Th2 response is necessary to protect a foetus from the rejection of the alloantigenic foetal structures by balancing the Th1 immune response. The predominance of the Th1 response has been associated with recurrent spontaneous abortion in women [13]. The Th1 response is essential for mounting a protective response against intracellular pathogens, including *C. abortus*, across host species both in vitro and in vivo [13,14,15] through the production of interferon-gamma (IFN-γ) [11,14,15,16]. Th17 cells produce interleukin (IL)-17A, which is involved in the immune regulation of extracellular microbes in different tissues, including the gastrointestinal tract, airways, mammary gland, uterus, and joints [11,17].

In contrast with what occurs in haemochorial placentation, in sheep, there is no immune modulation of the peripheral immune response [18]. When assessing systemic antigen-specific recall responses, no differences have been found in the IFN-γ, IL-4, or IL-10 production by peripheral blood mononuclear cells (PBMCs) between pregnant and non-pregnant sheep [18]. This may suggest that changes in the systemic balance between Th1 and Th2 cytokine production do not occur during pregnancy in sheep. It was hypothesised that due to the number of tissue layers between the foetal tissue and maternal blood, the protective pro-inflammatory maternal immune response might not reach the foetal tissues during pregnancy [19]. As a result, this could allow the invasion of certain abortifacient agents such as *C. abortus* [20], granting them access to epithelial trophoblast cells through the development of the haematoma in the placentome that is formed during the second month of gestation [21].

Additionally, it could be hypothesised that the immune response at the maternofoetal interface, including the maternal endometrium, could determine the outcome of pregnancy. The female genital tract also has a group of cellular immune components in the mucosa, including keratinocytes, antigen-presenting cells, and intraepithelial gamma-delta (γδ)-T cells. The γδ-T cells in the uterine epithelium of both pregnant and non-pregnant sheep have a fully differentiated effector phenotype that could, therefore, rapidly respond to any invading pathogen [22]. During parturition and puerperium, the uterus is particularly susceptible to infection from exposure to the vaginal flora and environmental contaminants, which coincide with a dramatic increase in the number and size of uterine γδ-T cells toward the end of pregnancy, which could facilitate the survival of the foetus in the delivery [23].

Since the immune response and hormonal balance at the maternofoetal interface change as pregnancy develops, it is important to define the immune composition following experimental *C. abortus* infection. The aim of this study was to assess the immune composition in the uteri and placentae of *C. abortus*-infected sheep, drawing associations between sheep with specific pregnancy outcomes. An improved understanding of the interactions between immune response in the maternofoetal interface and pregnancy outcomes may help to explain what triggers different pregnancy outcomes from EAE.

## 2. Materials and Methods

### 2.1. Animals and Experimental Design

The animals from which the samples in this study were selected originated from the unvaccinated challenge control and negative control groups of a vaccine-challenge trial, comparing and evaluating different antigen formulations for protective efficacy (unpublished). This study was carried out in strict accordance with the Animals Scientific Procedures Act 1986 and in compliance with all UK Home Office Inspectorate regulations. The experimental protocol was approved by the Moredun Experiments and Ethical Review Committee (Permit number: E30/17; approved on 20 March 2017). The challenge control and negative control animals were housed separately and in different buildings located away from each other on the farm. The challenge control animals were inoculated with 2 × 10^6^ inclusion forming units of the *C. abortus* strain S26/3 (genome accession number CR848038, [24]) by subcutaneous injection over the prefemoral lymph node at day 70 of gestation using our well-established and described pregnant sheep model [3,5,25,26]. These negative control animals were neither treated nor challenged. All animals were monitored throughout the study at least three times daily, receiving the appropriate veterinary care by veterinary practitioners according to UK animal welfare standards.

All animals were 3–6 years old Scotch Mule (crossbred sheep of Scottish Blackface ewes sired by Bluefaced Leicester rams) sheep from OEA-free flocks and seronegative for *C. abortus* by a rOMP90-3 indirect enzyme-linked immunosorbent assay [27]. The placental and uterine samples used in the present study were collected immediately after pre-term abortion/parturition. *C. abortus*-challenged ewes (*n* = 8) were selected and grouped according to their pregnancy outcome as Dead/Dead (DD) when both foetuses were aborted (*n* = 3); Dead/Live (DL) when one of the foetuses was aborted, and the other was born alive (*n* = 3); and Live/Live (LL) when both lambs were born alive (*n* = 2). Additionally, two non-challenged EAE-free pregnant ewes were selected as the negative controls (NC) where two twin lambs were born alive.

Following parturition, the sheep were euthanised with a jugular intravenous administration of 0.5 mL/kg pentobarbitone sodium solution (Pentoject 200 mg/mL, Animal Care Ltd., Newtown, UK). Immediately after the confirmation of death, samples of the uteri were collected.

### 2.2. Sample Collection and Storage

Placentae were collected at abortion/parturition, tagged, photographed, and macroscopically assessed to determine the extent of visible EAE lesions. The samples of two cotyledons, one proximal (Px) and one distal (D) to the umbilical blood vessels (Px ≤ 3 cm and D > 3 cm of distance), were collected from every placenta (P1, the first placenta found; P2, the second placenta found for each animal). Each of these cotyledons was divided into four parts: one part was frozen at −20 °C for downstream processing by a quantitative real-time PCR (qPCR) and for the preparation of smears for modified Ziehl-Neelsen staining (mZN); another was fixed in 4% paraformaldehyde–phosphate-buffered saline (PBS) solution (PFA) (*w*/*v*); another was fixed in a zinc salt fixative (ZSF) (pH = 7.4) [28]; while the remaining part was fixed in 10% neutral buffered formalin (NBF) (Cellstor Pot, CellPath Ltd., Newtown, UK). Two caruncle tissue samples (2.5 × 2 cm) from every uterine horn were also collected and treated as described above for the placentae (frozen at −20 °C or fixed in ZSF, 4% PFA, or 10% NBF). Cervicovaginal mucus (CVM) samples were collected by cervicovaginal swabs following the delivery of the placentae.

### 2.3. Real-Time PCR Analyses

The total DNA was extracted from 20 mg of the sampled material (placentae, uteri, or CVM) using a DNeasy Blood & Tissue kit (Qiagen Ltd., Crawley, UK), according to the manufacturer’s instructions. A quantitative PCR was carried out on the placental DNA samples using primers based on the major outer membrane protein (MOMP) gene (OmpA), as described previously [29]. Briefly, the PCR reaction consisted of a 2X TaqMan^®^ universal PCR master mix (Applied Biosystems, Warrington, UK), OmpA forward primer (5′-CGGCATTCAACCTCGTT-3′), a reverse primer (5′-CCTTGAGTGATGCCTACATTGG-3′), a fluorescent probe (TaqMan^®^probe, 5′ FAM-GTTAAAGGATCCTCCATAGCAGCTGATCAG-TAMRA 3′), 1 μL of DNA, and sterile nuclease-free water (Promega Corporation, Madison, WI, USA) up to a final volume of 25 μL per sample. The thermal cycling conditions were 50 °C for 2 min; 95 °C for 10 min; 95 °C for 15 s (45 cycles); and finally, 60 °C for 1 min. Amplification and detection were performed using a QuantStudio-5 real-time PCR System (Applied Biosystems, Foster City, CA, USA), following the manufacturer’s standard protocols. Each sample was tested in triplicate and quantified against a standard curve, which was established with 10-fold concentrations (ranging from 10^7^ to 10^1^ genome copies of *C. abortus* strain S26/3 per reaction) with nuclease-free water (Promega, Madison, WI, USA) as a negative control. The concentrations of placental DNA were determined using a NanoDrop ND-100 (NanoDrop Technologies, Wilmington, NC, USA) spectrophotometer, as previously published [29].

### 2.4. Phenotypic Analysis of Inflammatory Cells

To investigate immunopathology within the upper reproductive tract and the placentae of ewes following experimental *C. abortus* infection with different pregnancy outcomes, the presence of the *C. abortus* antigen was measured alongside the identification of the cellular phenotypes present in the inflammatory infiltrate by immunohistochemistry (IHC) and in situ hybridisation (ISH).

#### 2.4.1. Immunohistochemical Labelling of Tissue Sections

Uterine and placental samples were fixed in ZSF (for CD4^+^, CD8^+^, and γδ-T cells, NK cells, cytokines IL-10, and tumour necrosis factor-alpha (TNF-α) and *C. abortus* MOMP antigen) or 4% PFA (for forkhead box P3 (FoxP3)) and were embedded in paraffin wax to create paraffin blocks. The serial sections of each block were cut at 5 μm thickness and mounted onto positively charged adhesive slides (SuperFrost Ultra Plus™, Thermo Fisher Scientific, Braunschweig, Germany). Before starting, the slides were dried overnight at 37 °C. Every sample was identified using permanent markers (CellMark Pen, CellPath Ltd., Newtown, UK). Negative control sections were prepared for each sample substituting the primary monoclonal antibody (mAb) with an IgG isotype control diluted in TBS. Sections of the lymph node tissue were included in every run as positive controls. The slides were warmed up at 60 °C for 30 min and then dewaxed by xylene and rehydrated through graded alcohols (from 100 to 50%) and in distilled water using an automatic processor (Shandon Varistain, Thermo Fisher Scientific, Cheshire, UK).

#### 2.4.2. Pre-Treatment for Antigen Retrieval IHC

Fixed tissue sections on positively charged SuperFrost slides for the identification of FoxP3^+^ cells were placed in metal racks, submerged in a glass beaker containing a 0.21% citrate buffer solution (*w*/*v*, pH 6.0), and incubated in an autoclave at 121 °C for 10 min. After cooling to below 50 °C, the slides were washed in PBS and then in a 0.5% Tween 80-PBS solution (*v*/*v*). The slides were kept in PBS until processing for IHC.

#### 2.4.3. Immunolabelling of Tissue Sections

Tissue section slides were treated using a freshly made solution of 3% hydrogen peroxide (Perhydrol^®^, Merck Life Science UK Ltd., Gillingham, UK) in methanol (*v*/*v*) for 30 min to inactivate any endogenous peroxidase activity. The peroxidase solution was removed by washing it under running tap water for 5 min. The slides were placed in a staining chamber system (Sequenza^TM^, Thermo Fisher Scientific, Pittsburgh, PA, USA), washed with TRIS-buffered saline (TBS; Ph = 7.4) (Thermo Scientific Chemicals, Thermo Fisher Scientific, Lancashire, UK), and then treated with a 20% normal goat serum (Sigma-Aldrich, Merck Life Science UK Ltd., Gillingham, UK) in TBS (*v*/*v*) for 30 min to block the non-specific binding sites. Primary monoclonal antibodies diluted in TBS were added and incubated overnight at 4 °C (Table 1). After washing twice with TBS, a goat anti-mouse IgG conjugate (Envision™ + System horseradish peroxidase-labelled polymer, Dako, Ely, UK), 1/20 (*v*/*v*) was applied for 30 min according to the manufacturer’s instructions. After washing again with TBS, the slides were incubated with an aminoethyl carbazole alcohol soluble chromogen (AEC Red, Vector Laboratories, Cambridgeshire, UK) for 8 min before washing in tap water, counterstaining with haematoxylin (Hematoxylin Solution, Mayer’s, Merck Life Science UK Ltd., Gillingham, UK) and mounting with Histomount^TM^ (Thermo Fisher Scientific, Carlsbad, CA, USA). Positive and negative antigen controls were included in each run.

#### 2.4.4. Determination of the Optimal Concentration of Primary mAb for IHC

IHC markers (*n* = 8) were used to identify changes in immune cell infiltration in the uterus and placenta of pregnant sheep following experimental *C. abortus* infection. The fixative and IHC procedure was validated using ovine lymph nodes as a positive control for each round and for every marker used. In the case of IHC for IL-10 and FoxP3 and ISH for mRNA of IL-17A and IFN-γ, specific Chinese Hamster Ovary-transfected cell lines from in vitro culture were also used as positive controls with their derivation described elsewhere for continuous ovine cytokine [30] and FoxP3 transcription factor expression [31].

The specific mAbs used to detect CD markers (CD4+, CD8+, and γδ-T cells and NK cells), the transcription factor associated with Treg cells (FoxP3), cytokine antigens (IL-10 and TNF-α), and the chlamydial MOMP antigen (*C. abortus* MOMP mAb 4/11 was used as previously described [25,32,33]) are shown in Table 1. The optimal concentrations for each primary mAb were determined by increasing the dilutions (from 1/50 to 1/4000) of each mAb diluted in TBS. The highest dilutions where the cells were labelled without any background staining were selected. The *C. abortus* MOMP mAb was used at a dilution of 1/2000 (*v*/*v*) in TBS, which was tested using the *C. abortus* placentae as a positive control and *C. pecorum* infected fetal intestine samples as a negative control (Appendix A) [4,33,34].

**Table 1 pathogens-12-00846-t001:** Specific mAbs used for labelling different immune cell types and the *C. abortus* MOMP antigen in the ovine placentae and uteri.

Target	Antigen	Mab Clone	Dilution	Reference
Th cells	CD4	17D/2	1/400	[35]
T cytotoxic	CD8	SBU-T8	1/200	[35]
*C. abortus*	Chlamydial MOMP	4/11	1/2000	[33]
γδ-T cell	Gamma-delta T cell receptor (TCR)	86D	1/100	[36]
γδ-T cell	Gamma-delta workshop cluster-1 (WC1)	CC15	1/1000	[28]
NK cells	CD335/NKp46/NCR1	GR13.1	1/200	[37]
TNF-α	TNF-α	CC327	1/400	[32]
Treg cells	FoxP3	FJK-16s	1/200	[38]
IL-10	IL-10	CC318	1/100	[39]

#### 2.4.5. In Situ Hybridisation

Ovine (ov)IL-17A and ovIFN-γ mRNA were detected in 5 µm 4% PFA-fixed uterine and placental tissue sections through the identification of mRNA using an ACD RNAscope^®^ 2.5 HD Detection Reagent–RED kit (Advance Cell Diagnostics, Newark, NJ, USA), according to the manufacturer’s instructions. Briefly, after dewaxing in xylene and then in a methylated spirit at 99% (74 O.P.), the slides were air-dried and completely covered with a methylene RNAscope^TM^ hydrogen peroxide solution for 10 min at room temperature (RT), followed by washing in distilled water. Subsequently, the samples were subject to an antigen retrieval treatment using RNAscope^TM^ 1x retrieval reagents solution at 98–102 °C for 15 min. The solution was removed by immersion washing in distilled water and then in fresh 100% ethyl alcohol. The slides were completely air-dried at RT in a fume hood before being treated with RNAscope^TM^ protease plus at 40 °C for 30 min and washed in distilled water. The probe for ov-IL-17A [30] or ov-IFN-γ [40] (Table 2) was then added and incubated for 2 h at 40 °C in the hybridization oven (ACD HybEZ II, Biotechne, Minneapolis, MI, USA). An excess probe was removed by immersion washing in 1x RNAscope^TM^ Wash Buffer Reagents (RNAscope^TM^, Advance Cell Diagnostics, Newark, NJ, USA). Probe amplification was performed using the reagents AMP1 for 30 min at 40 °C, AMP 2 for 15 min at 40 °C, AMP 3 for 30 min at 40 °C, AMP4 for 15 min at 40 °C, AMP5 for 30 min at RT, and AMP6 for 15 min at RT. The slides were washed after each step with a 1x wash buffer. Finally, probe binding was revealed using the RNAscope^TM^ 2. 5 fast RED reagent (RNAscope^TM^, Advance Cell Diagnostics, Newark, NJ, USA) for 10 min at RT, followed by counterstaining with 50% Gill’s haematoxylin solution (Sigma-Aldrich, Gillingham, UK) (*v*/*v*) for 2 min at RT and consecutive washing in 0.02% ammonia water and tap water. After drying at 60 °C for 15 min, coverslips were placed over the tissue sections using a mounting solution (EcoMount mounting medium, Advance Cell Diagnostics, Newark, NJ, USA).

#### 2.4.6. Phenotypical Scoring

IHC labelled and ISH slides were digitised using a slide scanner (Nanozoomer XR, Hamamatsu, Photonics Ltd., Southampton, UK), as previously described [41]. Briefly, ten different randomly selected areas of 1 mm^2^ per slide were analysed, and the number of cells labelled in each region was counted using the pathology analysis software Qu-Path v0.3.0 [42]. For MOMP, the labelling was not scored by the number of cells but by the area labelled in relation to the total tissue present in each of the ten sections.

### 2.5. Statistical Analysis

The differences in lamb birth weights by the treatment group were assessed using an analysis of variance with *post hoc* comparisons calculated using Tukey’s Honest Significant Difference method at a 95% confidence level. Linear mixed modelling approaches were considered when accounting for the possible effect of a ewe; however, no variance was attributed to the ewe in the model.

The qPCR data from the uteri and placentae were measured as the number of genomic DNA (gDNA) copies and were analysed by fitting linear mixed models to the log of the responses with the group and organ (placentae or uteri) and the interaction between the group and organ included as fixed effects, where the possible groups were DD, DL, LL and NC, as described previously. Random intercepts were included in the model to account for the potential correlation between repeated measures from the same ewe and from the same placenta or horn of the uterus when nested within the ewe. qPCR data from the CVM swabs were analysed using linear models fitted to the log of the response with the group, which was included as a fixed effect. Random effects were excluded from the CVM analysis as there was only a single measurement per ewe.

The Immunolabelling counts were analysed using (zero-inflated) generalised linear mixed models (GLMMs) with a negative binomial response (with quadratic parameterisation), which was fitted to each of the 11 response variables and data from the uterus or placenta separately. The uterus and placenta were initially modelled separately to allow for substantial differences in the proportion of zero responses between these two organs. For all response variables, a simpler model without zero inflation was considered first; however, when model convergence issues or diagnostic plots revealed clear issues with the model fit due to a large number of zeros, a zero-inflated model was fitted. In cases where diagnostic plots for the non-zero-inflated model showed minor issues, the zero-inflated models were checked to ensure that the statistical significance of fixed effects was not unduly affected by the inclusion or exclusion of a zero component. The statistical modelling of immunolabelling data was not reported for response variables, which were too sparse for robust analysis. The group was included as a fixed effect in all models, as with the qPCR data, alongside random intercepts accounting for the ewe and the placenta or horn of the uterus nested within the ewe. When fitting zero-inflated models, the same set of random and fixed effects was considered for the zero parts: as in the non-zero part with terms dropped until a robust model could be fitted. The model effect sizes (es) were stated on the log scale used in the negative binomial model. All effect sizes and *p*-values relating to the differences between groups within an organ came from the models described above. The same approach was then conducted on the combined dataset containing both the uterus and placenta to assess if there were statistically significant differences between the organs, with the same random effects structures and with the addition of the organ and the interaction between the organ and group to the fixed effects.

For LMMs and GLMMs, the statistical significance of fixed effects was assessed at the 5% level using Wald chi-square tests. A false discovery rate method was used to account for the effects on multiple testing in comparisons with the groups for models with a statistically significant effect of the group. All analyses were performed on the R system for statistical computing [43] using the lme4 [44] and glmmTMB packages for model fitting [45], the car and emmeans packages for significance testing and comparisons between these groups [46,47], and the performance and DHARMa packages for model diagnostics [48,49].

## 3. Results

### 3.1. Clinical Outcomes and Group Allocation

The pregnancy outcome was determined at the time of abortion/parturition to allow for the allocation of animals into the different groups according to their clinical outcome. The two negative control ewes, NC (normal pregnancy), gave birth to four lambs (two sets of twins with a mean birth weight of 4.8 kg (5 and 4.5; 4.5 and 5.2 kg) at 147 and 147 days of gestation (dg), respectively). In group DD, three ewes gave birth to a total of six stillborn lambs (three sets of twins with a mean birth weight of 2.48 kg (2.1 and 2.2; 2.1 and 2.5; 2 and 4 kg) at 130, 133, 140 dg, respectively). In group DL, three ewes each delivered two lambs, one dead and one living (three sets of twins with a mean birth weight of 3.05 kg (2.3 and 3.6; 2.8 and 2.6; 3 and 4 kg) at 135, 140, 140 dg, respectively). In group LL, four lambs were born from the two ewes (two sets of twins with a mean birth weight of 3.58 kg (3.5 and 4; 3.3 and 3.5 kg); 140, 142 dg). There was a statistically significant effect of the group on the birth weights (*p* < 0.001). This difference was driven by a higher mean birth weight in the NC group than in any of the DD (*p* < 0.001), DL (*p* = 0.001), or LL (*p* = 0.030) groups. There were no statistically significant differences in birth weight between the three infected groups.

Placentae from the challenged sheep showed different levels of severity in gross pathology (Figure 1). In group DD, all the placentae showed gross pathological lesions, but two of them showed a high degree of autolysis (Figure 1B,C), making them difficult to analyse. In group DL, typical EAE lesions were also present, but in one of them, no gross lesions were found. In group LL, lesions were also present, and in one of them, the lesions were very severe (80%) (Figure 1 and Table 3). In the NC control group, no gross lesions were found in any of the four placentae.

### 3.2. MZN and qPCR Results in Placentae and Uteri

The presence of compatible chlamydial elementary bodies was confirmed in the uteri and placentae by mZN (Table 3). The estimation of bacterial load by the qPCR assay was assessed using a gDNA standard prepared from the *C. abortus* S26/3 strain. Testing was performed in triplicate, with the minimum limit of detection determined to be 10 gDNA copies per reaction. An analysis of the NC group samples yielded values between 0 and 9 gDNA copies in the uteri, 41 and 372 gDNA copies in the placentae, and 20 and 40 gDNA copies in CVM swabs. In samples from the uterus, placenta, and CVM swabs, there was a statistically significant effect demonstrated by the group in all cases (*p* < 0.001 for each of the uterus, placenta, and CVM swabs), and gDNA copies in the NC group were significantly lower than the responses of the challenge groups LL, LD, and DD (*p* < 0.001 in all cases).

In the uteri from infected sheep, the number of *C. abortus* gDNA copies ranged from 8.8 × 10^1^ to 3.2 × 10^5^, but no statistically significant difference between the challenge infection groups (DD, DL, and LL) were found (DD-DL *p* = 0.707, DD-LL *p* = 0.778, DL-LL *p* = 0.846). In the infected placentae, the range of observed numbers of *C. abortus* gDNA copies was slightly higher than in the uteri, which ranged from 1.3 × 10^3^ to 1.4 × 10^7^. No statistically significant effect of the group on the number of gDNA *C. abortus* copies present in the placenta was observed. There was some evidence of elevated numbers of gDNA copies in the LL group compared to the DD (*p* = 0.056) and DL (*p* = 0.051) groups, but this was marginally not statistically significant following its adjustment for multiple testing, and there was no evidence to suggest a difference between the DD and DL groups (*p* = 0.897). The number of gDNA copies was statistically and significantly higher in the placentae than in the uteri across all groups (*p* < 0.001). CVM swabs were taken following the delivery of the placentae, and these ranged from 1.8 × 10^5^ to 8.7 × 10^6^ gDNA copies when excluding the NC group. Similarly, to the results in the uteri, no statistically significant differences were found between the challenged groups in CVM (DD-DL *p* = 0.836; DD-LL, *p* = 0.335; DL-LL, *p* = 0.346).

### 3.3. Immunolabelling for C. abortus

In Section 3.3, Section 3.4 and Section 3.5, we present a discussion of the observed count data and the summary statistics stemming from these raw data before conducting a full statistical analysis of the data from Section 3.6 onwards.

The presence of *C. abortus* in the tissue sections was confirmed using a mAb (4/11) specific to *C. abortus* MOMP [33] (Figure 2 and Figure 3). No positive samples were found in the NC group. The median of the percentage for MOMP immunolabelling in groups DD, DL, and LL varied between 11.5 and 24.5% in the placentae and between 1 and 10% in the uteri (Appendix A and Table 4).

### 3.4. Phenotypical Determination of the Cell Infiltration

In the uteri (Figure 4), the positive labelling of immune markers was more abundant than in the placentae, with a high number for most of the markers investigated (Table 4). The median score was higher for DL ewes compared to the other infected groups for CD4 T cells, CD8 T cells, γδ-T cells, and TNF-α (Appendix A).

In the placentae, the infiltration of CD4 T cells, CD8 T cells, γδ-T cells, and NK cells were occasionally present in small numbers or in single cells (Figure 5 and Table 4). By contrast, immunolabelling for TNF-α, IL-10, and FoxP3 showed slightly higher cell counts in our samples (Figure 5 and Table 4). The detection of mRNA for IFN-γ and IL-17A was nearly absent in the NC tissues (Table 4).

### 3.5. In Situ Hybridization for IFN-γ and IL-17A mRNA

By ISH, IL-17A mRNA was detected at low levels in both the uteri and placentae in the DD group. The detection of mRNA for IFN-γ showed a higher presence in the uteri and placentae of DL and was only present at low levels in the DD and LL groups in the uteri (Table 4 and Figure 6 and Appendix A).

### 3.6. Statistical Analysis of the Immunolabelling and In Situ Hybridisation

Figure 7 shows the 11 variables (immune markers and chlamydial MOMP antigen), which are split by whether they were measured in the uterus or in the placenta. These plots suggest that, apart from MOMP, the counts were generally (though not always) lower in the placenta than in the uterus; however, there was rarely a clear difference between the groups, except for the NC group, which, as expected, often had lower counts.

### 3.7. Statistical Analysis of the C. abortus-MOMP Labelling

The NC group was excluded from the model comparing *C. abortus*-MOMP labelling percentages between the groups, as MOMP was never recorded in the NC group in either the uteri or the placentae. There was no statistically significant effect of the group on the response between the three infected groups (*p* = 0.186 in uteri and *p* = 0.186 in the placentae) (Figure 7 and Appendix A). The MOMP labelling was statistically significant and elevated in the placenta compared to the uterus (es = 1.01, *p* = 0.003).

### 3.8. Statistical Analysis of the Phenotypical Scoring in Uteri

There was a significant effect of the group for CD8 (*p* = 0.008), γδ-T cells (γδ-WC1, *p* = 0.016), and NK cells (*p* = 0.012), which was driven by the difference between DL and NC (es = 2.29, *p* = 0.005; es = 1.35, *p* = 0.008; and es = 1.74, *p* = 0.006, respectively).

An analysis of the CD4 T cells and γδ-T cells (γδ-TCR) labelling in the uteri showed similar but marginally non-significant effects of the group on this response (*p* = 0.057 and *p* = 0.059, respectively). In all cases, the estimated differences between the DD and LL groups and the NC group were in the same direction but with a slightly smaller magnitude, and so were not statistically significant; however, given the consistency of the estimated effects, it seems plausible that there could be a consistent and similar effect across the infected groups which this study was underpowered to detect.

The TNF-α labelling was very low in NC but elevated in the *C. abortus*-infected groups, showing the statistically significant effect of the groups (*p* = 0.030). However, no difference was found between the three infected groups (DD, DL, and LL). The statistically significant effect of the group was driven by the differences between the NC and DL (es = 2.64, *p* = 0.027) and between the NC and DD (es = 2.24, *p* = 0.047) groups. A similar but non-significant effect was also observed between NC and LL (es = 2.10, *p* = 0.078) (Figure 7 and Appendix A).

The IHC labelling suggested no difference between the groups for both IL-10 (*p* = 0.289) and Foxp3 (*p* = 0.966).

Using ISH, the counts of the labelled cells for IFN-γ were lower in the NC group than in the other groups (Appendix A). The very high proportion of zero values in the NC group created model fitting issues; therefore, this group was dropped from the model. A comparison of the infected groups showed no statistically significant effect of the group on the response (*p* = 0.663).

There was no evidence of a statistically significant difference between the groups for IL-17A (*p* = 0.072) (Figure 7 and Appendix A). A summary of the main findings is presented in Table 5.

### 3.9. Statistical Analysis of the Labelling in Placentae

The IHC labelling in the placenta showed a very high proportion of zero-labelling areas for CD8 (96% zeros) and γδ-T cells (using WC1 and γδ-TCR labelling) (91% zeros for each) and NK cells (94% zeros). Given that these high proportions of zeros posed model fitting issues even for zero-inflated models and that there was no suggestion of a difference between these groups in a visual examination of the data, these cell types were excluded from the statistical analysis.

Despite a relatively high proportion of zeros in the CD4 counts (86% zeros), a zero-inflated model could be fitted. This model showed no evidence to support a difference in the probability of a zero response between these groups; however, there was a statistically significant effect on the group for CD4 T cells in the non-zero part of the model (*p* < 0.001). Specifically, the mean non-zero response in the DD group was statistically and significantly higher than the mean non-zero response in any of the other three groups (DD-DL: es = 2.26, *p* < 0.001; DD-LL: es = 1.22, *p* = 0.016; DD-NC: es = 2.11, *p* = 0.002) (Appendix A); however, caution should be exercised in interpreting this result as it is based on a relatively small sample given the high proportion of zeros in the data.

There was no support for the difference in the probability of zero counts for TNF- α across the groups (*p* = 0.089); however, there was a statistically significant effect of the group on a non-zero part of the model (*p* = 0.002). Specifically, the mean non-zero response in the DD group was statistically and significantly higher than the mean non-zero response compared to any of the other three groups (DD-NC: es = 1.68, *p* = 0.001; DD-LL: es = 1.04, *p* = 0.037; DD-DL: es = 1.18, *p* = 0.015) (Figure 7 and Appendix A). 

When considering FoxP3 labelling, there was support for a statistically significant difference in the probability of zero values across the groups (*p* = 0.010) and for the statistically significant effect of the group on the non-zero part of the model (*p* < 0.001). Following an adjustment for multiple testing, there was a statistically significant larger proportion of zero counts in the DD group than in the DL group (es = 4.02, *p* = 0.013). There were also estimated differences in the same direction but of a slightly smaller magnitude between the DD group and both the LL (es = 2.29, *p* = 0.159) and NC (es = 3.60, *p* = 0.053) groups which were not statistically significant. In the non-zero part of the model, the counts in the DD group were statistically significantly lower than those in the DL (es = 1.48, *p* < 0.001) and LL (es = 0.89, *p* = 0.041) groups, with a similar slightly smaller non-significant difference with the NC group (es = 0.62, *p* = 0.100). Finally, the counts in the DL group were significantly higher than those found in the NC group (es = 0.86, *p* = 0.009).

For IL-10, there was no support for a difference in the probability of zero values across the groups; however, there was a statistically significant effect of the group on the non-zero part of the model (*p* < 0.001). Specifically, the mean non-zero response in the NC group was statistically and significantly lower than the mean non-zero response in any of the three infected groups (DD-NC: es = 1.42, *p* < 0.001; DL-NC: es = 1.67, *p* < 0.001; LL-NC: es = 1.54, *p* < 0.001).

In situ, hybridization labelling for IFN-γ and IL-17A in the placenta also showed a great proportion of zeros with marginally statistically significant effects of the group on the non-zero part of each model (IFN-γ *p* = 0.022 and IL-17A *p* = 0.032), though the only significant difference between the groups was a higher non-zero response in the DL group than the NC group for IFN-γ (es = 1.41, *p* = 0.018) (Figure 7 and Appendix A).

### 3.10. Statistical Comparison of the Labelling between the Uteri and the Placentae

The cell counts were statistically and significantly higher in the uterus than the placenta for CD4 (es = 4.74, *p* < 0.001), CD8 (es = 6.47, *p* < 0.001), γδ-T WC1 (es = 4.73, *p* < 0.001), TNF- α (es = 0.88, *p* = 0.035), IL-10 (es = 2.31, *p* < 0.001), FoxP3 (es = 1.1, *p* < 0.001) and IFN-γ (es = 1.17, *p* = 0.001). There was also evidence of a statistically significantly higher proportion of zero values in the placentae than the uteri in γδ-T TCR (es = 5.63, *p* < 0.001) and NKp46 (es = 4.48, *p* < 0.001), alongside higher non-zero counts in the uterus than the placentae in γδ-T TCR (es = 2.40, *p* = 0.015) and NKp46 (es = 2.96, *p* < 0.001). There was no evidence of a statistically significant difference in IL-17A counts between the uteri and the placentae. Robust comparisons between the uteri and placentae are often challenging due to sizeable differences in the proportion of zeros, which makes model fitting difficult and occasionally sensitive to small changes in the model specification; therefore, these results should be interpreted with caution, particularly when effect sizes are small, and *p*-values are close to the threshold for statistical significance.

## 4. Discussion

EAE is a systemic disease of ewes, which are infected through the oronasal mucosa resulting in the infection of the reproductive unit comprising the uterus, placenta, and foetus, and resulting in abortion in the last few weeks of pregnancy. The role of some immune cells and molecules has not been established in sheep reproductive tissues, in part due to a historical lack of specific validated reagents [50]. This study aimed to further elucidate the processes that occur during the invasion of *C. abortus* and the accompanying immune cell infiltrate to maternal endometrial tissues and foetal placental tissues following experimental *C. abortus* infection (modelling natural disease) and pre-term abortion and/or lambing. The aim was to link the patterns of cellular immune infiltration in the endometrial tissues and the placentae with the specific outcomes of pregnancy. To achieve this, we used maternal uterine tissues and the placentae collected from ewes-bearing twins at parturition where both lambs were aborted (DD) or born alive (LL), or a mix of the two were delivered (DL) and made direct comparisons with uninfected controls (NC).

The uterus is the maternal organ that contains the conceptus (placenta and foetal tissues) between implantation and parturition. This organ provides nutritional, hormonal, and immune support to the development of the foetus and through to parturition. However, immunomodulation at the maternofoetal interface is essential for maintaining pregnancy through tolerance to allogenic foetal tissues, and the interference of this through infection is likely to influence pregnancy outcomes. The exchange and interaction between the mother and foetus are through the placenta that implants into the uterus. The placenta is a functioning hybrid organ that is formed by an apposition of maternal and foetal tissue that regulates the physiological, hormonal, and immunological interactions. The number of layers of tissues between the foetal and maternal blood is variable between mammalian species, where placentae are classified according to how many layers of maternal tissue are removed during chorion development [51]. It is hypothesized that there may be different requirements of the immune response to support the maintenance of pregnancy across mammalian species of differing placentation types [10]. This immune response could vary in response to the infections targeting these tissues, such as EAE; the focus of this study, which is on the systemic disease of ewes infected through the oronasal mucosa, results in the infection of the reproductive unit comprising the uterus, placenta, and foetus and causing an abortion in the last few weeks of pregnancy. Thus, this study aimed to describe the invasion of *C. abortus* and immune cell infiltrate to maternal endometrial tissues and foetal placental tissues following experimental *C. abortus* infection (modelling natural disease) in an attempt to link this with the specific outcomes of pregnancy in terms of abortion and/or live births. As part of this work, the role of some immune cells and molecules, such as FoxP3, IL-10, and IL-17A, which have so far not been established in sheep reproductive tissues, in part due to a historical lack of specific validated reagents, was described [50]. The immune responses of these tissues were evaluated by a determination of the cell surface antigens of selected immune cells (CD4, CD8, γδ-T cells, and NK cells), the intranuclear transcription factor, FoxP3 (Treg cells) alongside specific cytokines that are associated with inflammation and its control (IFN-γ, TNF-α, IL-17A, and IL-10).

In the placentae, gross lesions were variable, with no observable relationship to pregnancy outcomes, e.g., a placenta with a lesion affecting 20% of the total area in the DD group where its corresponding foetus was aborted was compared to lesions of 80 and 40% in the LL group placentae where all lambs were born alive. We observed that the estimated mean birth weight was lowest in group DD (mean 2.48 Kg), slightly increased in group DL (mean 3.05 Kg), increased further in group LL (mean 3.58Kg), and highest in group NC (mean 4.8 Kg); however, this was the only statistically significant difference between the NC group and each of the three infected groups. This would be expected because, in the infected ewes, the placentae had lesions that may have affected foetal nutrition and, therefore, could affect foetal development, as previously reported [3].

In addition, immunohistochemistry labelling for *C. abortus* was performed using a mAb that is specific for detecting *C. abortus*-MOMP rather than one based on the genus-specific chlamydial LPS as mAb 13/4 [33,52]. The specificity of mAb for *C. abortus* MOMP ensured that it was specific for *C. abortus* and not for other bacteria from the genus *Chlamydia*. In general, the immunolabelling for MOMP showed less chlamydial antigen in the uterine tissues than in the placental tissues. Similarly, the mean number of *C. abortus* gDNA copies appeared to be higher in the placentae than in the uteri. This difference in the presence of the organism between the tissues might be associated with the immune response controlling proliferation in the uterine tissues, but due to the type of placentation, in part, this immune response would be insufficient to control the disease developing in the foetal placenta. In the case of *C. abortus*-infected animals, the maternofoetal interface immunity seemed to play an essential role in maintaining the balance between controlling infection and preserving the pregnancy.

Although chlamydial organisms were observed in the positive groups (DD, DL, and LL), LL group placentae showed the highest immunolabelling for MOMP. Overall, for qPCR, LL placentae had the highest organism load, while NC had the lowest. An examination of the raw data in the uteri also revealed that although gross lesions were of greater severity in the horns of DD than in LL, more chlamydiae were present in LL than in DD. This could indicate that chlamydiae have recently entered the trophoblast cells and have not fully destroyed the integrity of tissues and appear to remain actively replicating in the intact trophoblast layer, which is no longer present in the aborted conceptus (of the DD group). This could be due to the ischemic autolysis caused by disconnections with the maternal tissue and, hence, with the nutrient supply. These results are suggestive of differences between the pregnancy outcomes of infected sheep, but it should also be remembered that these observations are only from two to three animals per group using small randomly selected samples of the uteri and the placentae, and so it would be worth examining this further with larger numbers.

In contrast to the foetal placentae, where a large number of zeros make it difficult, in some cases, to perform robust statistical analysis, the immune composition at the maternal uterus showed statistically significant differences between groups. In the uteri, there were differences in some immune targets when measured between uninfected (NC) and *C. abortus*-infected (DD, DL, and LL) sheep. CD8, γδ-T cells (positive for γδ-TCR and γδ-WC1), NKp46, TNF-α, FoxP3, and IL-10 cell labelling were significantly lower in NC than in DL, and the estimated differences between the NC group and the three infection groups were generally similar, though not consistently statistically significant. As mentioned previously, this could be associated with a barrier resulting from the number of layers separating maternal and foetal blood (synepitheliochorial placentation).

CD8 T lymphocytes have been reported as the predominant lymphoid cell type infiltrating the genital tract of primates infected with *C. trachomatis* [53], playing an essential role in *C. abortus* infection clearance [54] and abortion in mice and sheep [19,55]. In this study, elevated CD8 T cells were found in the uteri of the mixed pregnancy outcome group DL relative to the uninfected NC group. This could be related to bacterial control but also to a failure to effectively control and protect from abortion, as previously described [25,56]. Although the number of animals in each group was certainly a limiting factor when reaching conclusions, the mixed outcome observed in DL sheep could suggest an additional immune stimulation due to the presence of a dead foetus. Furthermore, the retention of the conceptus in the uterus for a longer period would provide more time for the live foetus to develop an immune response. It may promote a higher lymphoid infiltration from the maternal side, which may not be physiologically necessary for DD, where both foetuses were dead and had probably lost contact with maternal tissues.

γδ-T cells are also one of the most important populations of uterine intraepithelial lymphoid cells in many species, including humans, ruminants, and mice [57]. After parturition, these cells increase in number for the control of intra-vaginal infections caused by pathogens that could infect the uterus [23,58]. An elevation in the amount of these cells was also found in cases of pyometra in canines [59]. In this study, the number of γδ-T cells showed a statistically significant difference only between the infected DL group and the uninfected NC group, suggesting that the *C. abortus* infection could induce the recruitment of this type of cell in uterine tissues [55,60]. This fact could be related to local suppurative inflammation, where polymorphonuclear cells (PMNs) can produce an additional stimulus to recruit γδ-T cells, resulting in a local increase in the number of these cells compared with the chlamydial-free NC group where the PMNs are not so abundant.

NK cells are also important in host defence against infectious agents by regulating the development of both innate and adaptive immune responses but also have prominent roles in decidual remodelling in species with haemochorial placentation [37]. In humans and mice, the uterine NK granulated cells contain perforin and possess cytotoxic activity toward the trophoblast in vitro [61], and the activation of NK cells by polynucleotides can cause abortion in pregnant mice [62]. Similar functionality has been observed in sheep following implantation, suggesting their role in immune modulation at the maternofoetal interface [37,63,64]. It is likely that the expression of these cells is temporal, as they are not found in the placenta at term [10] and could be initiated by local infections. NK cells also play an important role in the initial stage of *C. abortus* infection and the destruction of trophoblast *C. abortus*-infected cells in mice [19,26,65]. In this study, there was higher NK labelling in the DL group compared to the NC group in the uteri (but no difference between the infected groups), which could be associated with contributing toward the abortion. This could be due to the possible stimulation of local immune responses to the presence and growth of chlamydiae, causing trophoblast cell lysis and subsequent abortion. In contrast, NK cells were almost absent in the uninfected (NC) group, which is consistent with observations from previous reports [10,30]. By contrast, analysis of Treg-associated FoxP3 and IL-10 and inflammatory cytokines IFN-γ and TNF-α revealed greater frequencies in cell labelling in the uteri rather than the placentae, which are consistent with cell surface marker labelling.

IFN-γ has been reported as a necessary component for controlling chlamydial growth but has been consistently shown to be incompatible at key stages during successful pregnancies at the maternofoetal interface [20,66]. Therefore, it is conceivable that increases in the level of IFN-γ could occur in *C. abortus*-infected animals at parturition (DD, DL, and LL) and not in uninfected ones (NC) in accordance with previous reports [15]. Surprisingly, no difference was found among the *C. abortus*-infected groups, which showed different pregnancy outcomes. Differences in the labelling between the infected and NC group could indicate that this response may be associated with infection and may have a non-essential role in contributing to abortion in this natural host species; however, due to the large number of zero labelling, it is necessary to use a larger number of animals to support these findings.

The magnitude of TNF-α labelling in non-zero cell counts was higher and showed statistically significant differences between the infected groups in the placenta (*p* = 0.002) but not in the uteri. As was mentioned previously, TNF-α may cause the typical thrombosis observed in the placenta. This cytokine is mainly produced by local macrophages and results in the activation of NK cells, which stimulate macrophage activity, creating a vicious circle that could potentially end in foetal death [19]. Consequently, the presence of higher TNF-α levels in the DD placentae is not surprising, demonstrating a relationship between this cytokine and abortion in sheep. Within the placenta, foetal trophoblasts are now considered to be a major source of this cytokine (from in vitro and ex-vivo *C. abortus* infections), contributing to the inflammatory cascade and leading to the destruction observed at the placenta and abortion [32].

In the case of CD4, FoxP3, IL-10, and IL-17A in the uteri, there were no statistically significant group effects. In species with haemochorial placentation, such as mice, rats, and humans, immune regulation (systemic and local to the placenta) is essential to ensure the survival of the semi-allogeneic foetus, including increases in Treg/IL-10 [14]. However, in sheep, investigations into pregnant sheep have focussed on the analysis of systemic PBMC, where no differences in the antigen-specific peripheral immune responses between pregnant and non-pregnant ewes have been observed [14]. For the first time, FoxP3, which is normally present in the maternofoetal interface of species with haemochorial placentation, was also identified in NC uteri tissues in this study. FoxP3 labelling is unique to Treg cells (the main source of IL-10), which, in this study, were low and showed no difference between the infected groups in the uteri. It suggests that FoxP3 and IL-10 may have a role in pregnancy but not necessarily in *C. abortus* pathogenesis in the uteri (*p* = 0.965 and *p* = 0.289, respectively). By contrast, in the placenta, Foxp3, CD4, and IL-10 labelling showed statistically significant differences (*p* < 0.001). Specifically, DD was statistically lower than DL and LL, while NC was significantly lower than DL in the non-zero part of the model for FoxP3, and CD4 was statistically and significantly higher in DD than in the other groups in the non-zero part of the model. IL-10 counts were statistically and significantly higher in the three infected groups than in the NC group. These results may Indicate a higher Treg infiltration into the foetal placenta from animals with infected and live foetuses, which could support their progression through to parturition. However, due to the high number of zero labelling in the cytokines and transcription factor molecule results overall (IFN-γ, TNF-α, IL-10, and FoxP3), these observations should be carefully taken.

Th17 cells play an important role in the control of extracellular bacterial and fungal infections, mainly in the respiratory and gastrointestinal tracts but are also associated with chronic inflammation and have an unclear role in pregnancy [30,67,68,69]. In this study, IL-17A labelling was barely present in the uteri and in the placentae of all infected groups, with no clear associations with disease or pregnancy.

An increase in uterine intraepithelial lymphoid infiltration has also been reported as a normal response prior to parturition in order to aid in the control of intrauterine infections [23,58]. In this study, this infiltration was predominantly observed in the uteri. In the sheep placentae analysed in this study, γδ-T cells were found in small numbers or as single cells, appearing slightly more numerous in the mesenchyme than in the chorionic epithelium and were also found to infiltrate into subendothelial and mural (tunica intima, media, and adventitia) cells in cases of arteriolitis or arteritis, as previously reported [25]. This difference found between mice and sheep could be related to differences in their placentation type (haemochorial and synepitheliochorial, respectively), where the possibility of cellular migration from maternal blood may be different. In this study, two different mAbs that target different components of γδ-T cells (γδ-TCR and γδ-WC1) were used. However, although there was a slight difference, the labelling of γδ-T cells in both mAb was as single isolated cells or in the absence of labelling, supporting these previous findings [25]. In other mammalian species, such as mice and humans, maternal immunity plays an important role in protecting the conceptus. In this study, the low frequency of these cells present in the sheep placentae of diseased animals may suggest the irrelevant role of γδ-T cells when controlling infection in tissues at term.

Lymphoid infiltration in the placentae was remarkably lower than in the uteri, showing a great proportion of tissues with very few or no cells present (for CD4, CD8, γδ-T, and NK cells). Although placentitis in EAE is mainly suppurative with a high proportion of PMNs present, which is associated with the recruitment of lymphocytes, especially CD8^+^T cells [54], the number of these lymphocytes infiltrating the placenta was low, as in agreement with previous reports [25,56].

## 5. Conclusions

The present study provides novel detailed information about the immune responses at the maternofoetal interface in sheep, complementing and extending previous studies. The use of a panel of ten different immune markers with one chlamydial antigen marker in the same sheep tissue alongside the cotyledons and caruncles used in this study allowed a detailed comparison of the results with those of previous studies. This study also established, for the first time, a benchmark for comparing the presence of cells and cytokines in sheep reproductive tissues at the parturition and added new information on the presence of specific cytokines and cells, which could be associated with different pregnancy outcomes in *C. abortus*-infected ewes. These results revealed that the presence of specific cytokines and cells could be related to different outcomes, particularly in uteri tissues rather than the placentae. The role of regulatory cell activities, including IL-10 and FoxP3 expression, could be more nuanced, and further information is needed to evaluate their role in the maternal response to *C. abortus* infection in sheep. The levels of inflammatory cytokines such as TNF-α or the presence of NK cells could also be associated with abortion resulting from *C. abortus* infection. It would be interesting to explore the role of the immune response in the uterine tissue further to improve our understanding of its role in EAE, perhaps through the use of an in vitro organoid culture.

Although the results of this study show some associations, the intrinsic variation between animals and the low number of animals per group mean that it is difficult to determine any conclusive associations between different pregnancy outcomes in *C. abortus*-infected ewes. Future studies that involve larger sample sizes to strengthen evidence of current observations or a greater depth of analysis through the incorporation of new gene profiling in combination with antibody cell identification with spatial transcriptomics could enable the differential identification of gene pathways that are unique to pregnancy outcomes following experimental *C. abortus* infections.

## Figures and Tables

**Figure 1 pathogens-12-00846-f001:**
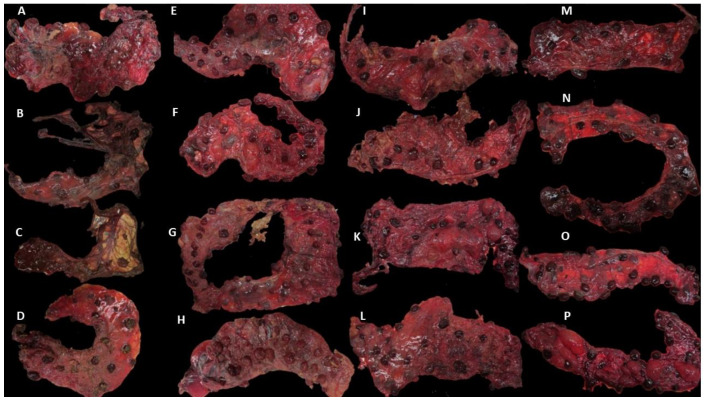
Placentae from the different clinical outcome groups. Placentae were identified by the number of the sheep, group (DD: dead/dead; DL: dead/live; LL: live/live; NC: negative control), and the number of placentae collected from each sheep (P1 and P2). Placentae showed different degrees of gross lesions (extent of lesions expressed as a percentage of placental surface area, shown in brackets): (**A**) Ewe 1 P1 DD (60%); (**B**) Ewe 2 P1 DD (100%); (**C**) Ewe 2 P2 DD (100%); (**D**) Ewe 3 P1 DD (50%); (**E**) Ewe 4 P1 DL (90%); (**F**) Ewe 4 P2 DL (0%); (**G**) Ewe 5 DL (fused placentae) P1 (90%) and P2 (25%); (**H**) Ewe 6 P1 DL (100%); (**I**) Ewe 7 P1 LL (80%); (**J**) Ewe 7 P2 LL (40%); (**K**) Ewe 8 P1 LL (10%); (**L**) Ewe 8 P2 LL (40%); (**M**) Ewe 9 P1 NC (0%); (**N**) Ewe 9 P2 NC (0%); (**O**) Ewe 10 P1 NC (0%); (**P**) Ewe 10 P2 NC (0%).

**Figure 2 pathogens-12-00846-f002:**
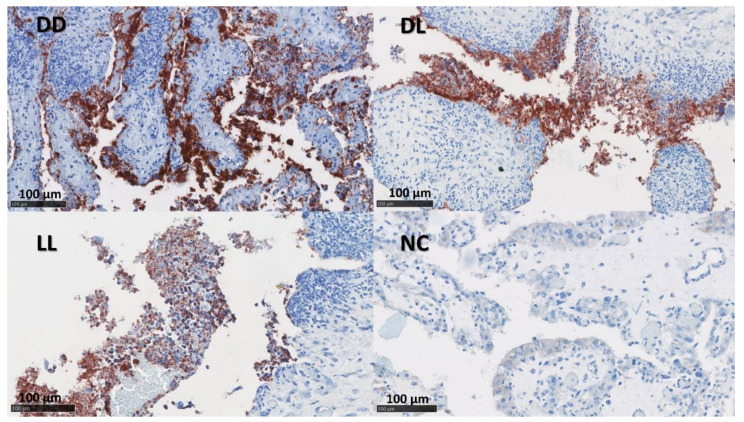
IHC labelling for *C. abortus* in the placenta. The figure depicts immunolabelling for *C. abortus* in the placenta using mAb 4/11 targeting *C. abortus*-MOMP in the experimental groups: Dead/Dead (DD), Dead/Live (DL), Live/Live (LL), and Negative Control (NC). *C. abortus*-MOMP is labelled in dark red over a blue counterstain. IHC used the AEC red substrate and was counterstained with haematoxylin.

**Figure 3 pathogens-12-00846-f003:**
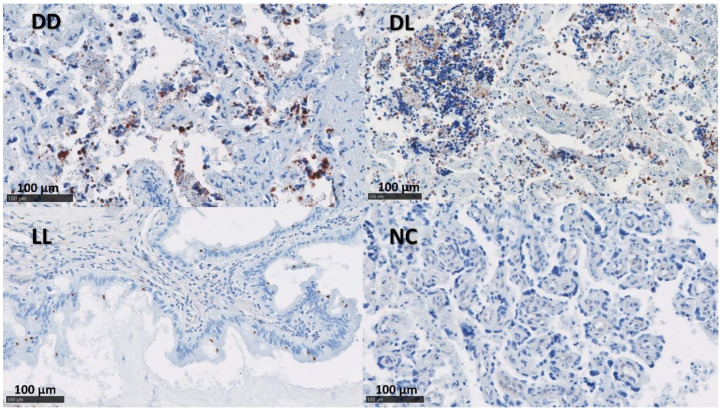
IHC labelling for *C. abortus* in the uteri. The figure shows immunolabelling for *C. abortus* in the placenta using mAb 4/11 targeting *C. abortus*-MOMP in the experimental groups: Dead/Dead (DD), Dead/Live (DL), Live/Live (LL), and Negative Control (NC). *C. abortus*-MOMP is labelled in dark red over a blue counterstain. IHC used the AEC red substrate and counterstained it with haematoxylin.

**Figure 4 pathogens-12-00846-f004:**
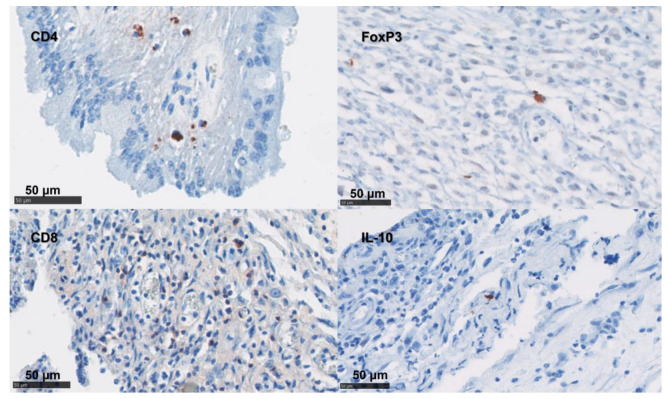
Labelling of ovine uteri using IHC for CD Markers. IHC using mAb targeted CD4 T cells, FoxP3 cells, CD8 T cells, and IL-10 cells were labelled with AEC red (Vector Laboratories, Peterborough, UK) over a blue counterstain with haematoxylin.

**Figure 5 pathogens-12-00846-f005:**
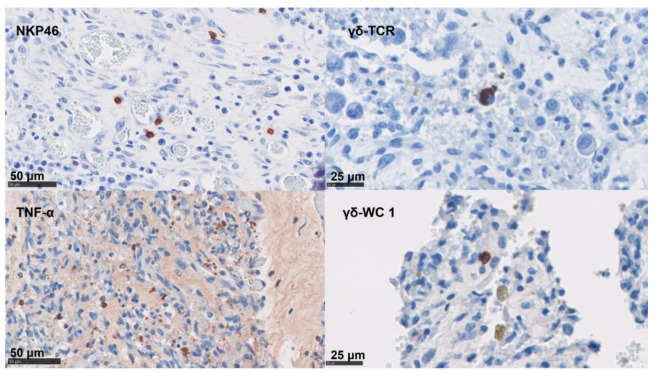
IHC labelling for lymphoid cells in the placenta. IHC using mAb-targeting NK cells, the γδ- T cell (TCR and WC1 antigen), and TNF-α; these are labelled with AEC red (Vector Laboratories, Peterborough, UK) over a blue counterstain with haematoxylin.

**Figure 6 pathogens-12-00846-f006:**
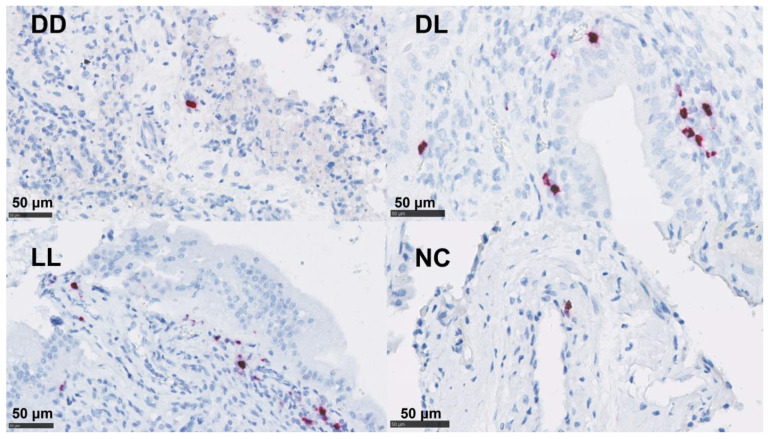
mRNA detection of IFN-γ by ISH in the placenta. Dead/Dead (DD), Dead/Live (DL), Live/Live (LL), and Negative Control (NC). ACD RNAscope^TM^ RED kit (Biotechne, Minneapolis, MI, USA).

**Figure 7 pathogens-12-00846-f007:**
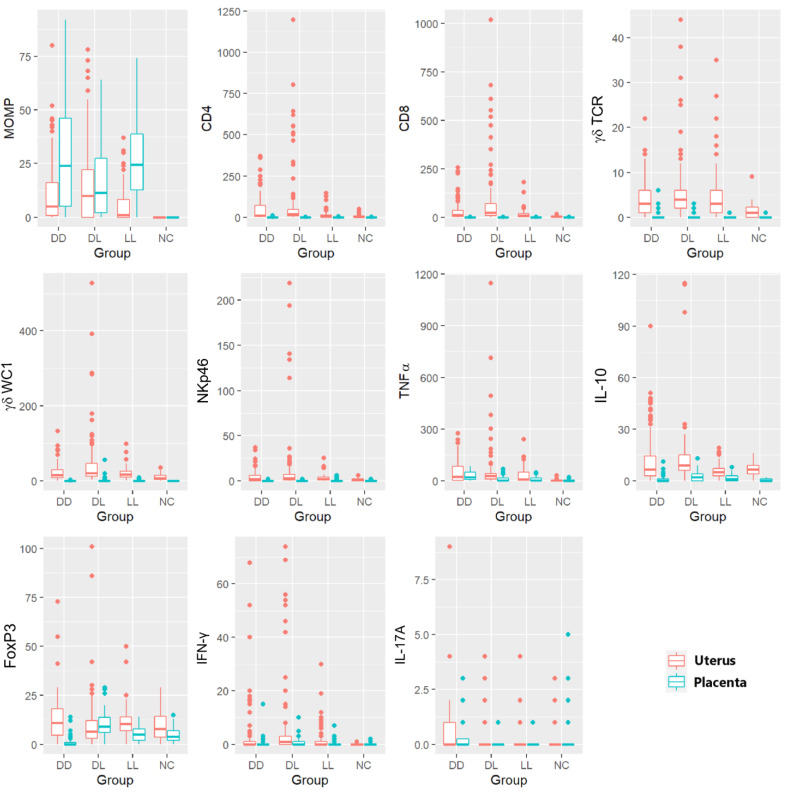
Box and whisker plots showing the multiple comparisons of immune markers and the *C. abortus* marker in the uteri and placentae. Different markers for IHC (*C. abortus* MOMP, CD4 T cells, CD8 T cells, γδ-T cells (γδ-TCR and γδ-WC1), NKp46, TNF-α, IL-10, FoxP3) and ISH (IFN-γ and IL-17A) in different ewe outcome groups (DD, DL, LL, and NC). Every boxplot identifies the uteri in orange and the corresponding placentae in light blue for each group of animals. Boxes in the plots are shown using the first quartile (bottom line of the box), median (central line in the box), and third quartile (upper line of the box). The whiskers (vertical lines) show a measure in the range, which could be extended to the most extreme data point no more than 1.5 times the interquartile range from the box. Outlier data are shown as solid points.

**Table 2 pathogens-12-00846-t002:** Probes used for the identification of cytokines by in situ hybridization.

Probe	Reactivity	NCBI Accession No	Reference
RNAscope^TM^ Probe-Oa-IL17A-C1	IL-17A	XM_004018887.4	[30]
RNAscope^TM^ Probe-Oa-IFNG-XCh	IFN-γ	NM_001009803.1	[40]

**Table 3 pathogens-12-00846-t003:** Summary of the results by ewe outcome group (DD, DL, LL, and NC) for gross uterine and placental pathology scoring and chlamydial load in the placentae (mZN staining and qPCR). A proximal (Px) and distal (D) cotyledon from each placenta (P1 and P2); two caruncles (C1 and C2) from each uterine horn (H1 and H2) and CVM swabs (Swabs) were sampled for each sheep. Macroscopic lesions were classified as the percentage of gross lesions covering the total surface area of the placenta/uterus. MZN scores were determined following the staining of the smears of every sample using a 4-degree scale: (negative (−); the occasional presence of elementary bodies (+); moderate (++); high (+++)). The number of genomic copies of *C. abortus* was also evaluated by the qPCR of extracted DNA in the same tissue samples (qPCR). NF, placenta not found. NA, not applicable.

Animal ID	No	1	2	3	4	5	6	7	8	9	10
	Group	DD	DD	DD	DL	DL	DL	LL	LL	NC	NC
Macroscopic lesions (%)	H1	90	100	100	70	30	100	20	10	0	0
	H2	25	20	100	30	90	100	10	40	0	0
mZN	H1C1	+++	++	+	++	+	++	+++	++	−	−
	H1C2	++	++	+++	++	+	++	+++	++	−	−
	H2C1	++	+	++	+	++	+++	+++	++	−	−
	H2C2	++	+	++	+	++	+++	+++	++	−	−
qPCR	H1C1	2.10 × 10^4^	1.70 × 10^4^	1.30 × 10^3^	1.70 × 10^4^	2.10 × 10^3^	1.70 × 10^4^	3.20 × 10^5^	1.20 × 10^4^	0	2.7
	H1C2	3.00 × 10^3^	1.10 × 10^4^	1.00 × 10^4^	5.90 × 10^3^	2.70 × 10^3^	2.80 × 10^4^	1.60 × 10^2^	3.50 × 10^4^	2	1
	H2C2	8.30 × 10^3^	5.90 × 10^2^	9.20 × 10^3^	1.90 × 10^3^	5.60 × 10^4^	1.70 × 10^5^	5.10 × 10^4^	1.00 × 10^4^	1.5	8.8
	H2C2	5.50 × 10^4^	8.8	4.20 × 10^4^	1.10 × 10^3^	5.70 × 10^4^	1.00 × 10^5^	1.30 × 10^5^	3.70 × 10^3^	8.5	3.2
Macroscopic lesions (%)	P1	60	20	80	40	90	NF	80	10	0	0
	P2	NF	40	NF	0	25	100	20	40	0	0
mZN	P1 Px	++	++	++	++	+++	NF	+++	+++	−	−
	P1d	++	+	+	++	+++	NF	++	+++	−	−
	P2 Px	NF	+++	NF	+	++	++	+++	+++	−	−
	P2D	NF	+++	NF	+	+++	++	+++	+++	−	−
qPCR	P1Px	2.10 × 10^5^	9.00 × 10^3^	4.30 × 10^6^	3.00 × 10^5^	2.00 × 10^6^	NF	5.50 × 10^6^	1.40 × 10^7^	170	45
	P1D	6.40 × 10^5^	3.90 × 10^3^	2.30 × 10^6^	1.00 × 10^5^	2.00 × 10^6^	NF	5.60 × 10^6^	1.10 × 10^7^	120	41
	P2Px	NA	2.30 × 10^6^	NA	1.30 × 10^3^	2.40 × 10^5^	1.00 × 10^7^	1.20 × 10^7^	2.40 × 10^6^	370	48
	P2D	NA	8.80 × 10^5^	NA	9.10 × 10^4^	2.70 × 10^4^	3.20 × 10^6^	8.50 × 10^6^	1.10 × 10^7^	170	41
Swabs		1.80 × 10^6^	3.50 × 10^6^	4.30 × 10^6^	6.50 × 10^5^	2.50 × 10^6^	8.70 × 10^6^	1.80 × 10^5^	2.90 × 10^6^	40	20

**Table 4 pathogens-12-00846-t004:** Summary statistics of the score of IHC markers for both the uteri and placentae sorted by group as either Dead/Dead (DD), Dead/Live (DL), Live/Live (LL), and Negative Control (NC). The markers were previously described (see Table 1). Q1, Med, and Q3 indicate the first quartile, median, and third quartiles of the datasets, respectively.

Tissue		Uteri				Placentae			
Group		DD	DL	LL	NC	DD	DL	LL	NC
MOMP	Q1	0.75	0	0	0	5	2	12.75	0
	MED	5	10	1	0	24	11.5	24.5	0
	Q3	16	22	8.3	0	46.25	27.5	38.75	0
CD4	Q1	3.75	7.25	3	1	0	0	0	0
	MED	11	19	6	3	0	0	0	0
	Q3	70.25	47.75	14.25	6.5	0	0	0	0
CD8	Q1	3.75	9.25	3	1	0	0	0	0
	MED	11.5	22.5	8.5	4	0	0	0	0
	Q3	35	71.5	19	6.25	0	0	0	0
γδ-TCR	Q1	1	2	1	0	0	0	0	0
	MED	3	4	3	1	0	0	0	0
	Q3	6	6	6	2.25	0	0	0	0
γδ-WC1	Q1	7.75	12	10	3.75	0	0	0	0
	MED	16	20	17	7.5	0	0	0	0
	Q3	29.5	45.75	26	15	0	1	0	0
NKp46	Q1	0	1	1	0	0	0	0	0
	MED	2	3	2	1	0	0	0	0
	Q3	6	7	4	2	0	0	0	0
TNF-α	Q1	2	11.3	2	0	8	0	0	0
	MED	24	28	8.5	1	20.5	3	2	0
	Q3	85	41	50	4	51	15.25	15	5
IL-10	Q1	3	6	2.75	4	0	0	0	0
	MED	6.5	9	5	6.5	0	2	1	0
	Q3	14.25	15	7.25	9	1	4	3	1
FoxP3	Q1	4.75	3	7	3.75	0	6	2	2
	MED	11	6.5	10.5	8	0	9	5	4
	Q3	18.25	12	14	14.25	1	13.75	8	7
IFN-γ	Q1	0	0	0	0	0	0	0	0
	MED	0	1	0	0	0	0	0	0
	Q3	1	3	1	0	0	1	0	0
IL-17A	Q1	0	0	0	0	0	0	0	0
	MED	0	0	0	0	0	0	0	0
	Q3	1	0	0	0	0.25	0	0	0

**Table 5 pathogens-12-00846-t005:** This table shows the estimated mean counts from the IHC and ISH based on back-transformed estimates from the (non-zero parts of the) negative binomial generalised linear mixed models. Estimated counts are shown as the means with 95% confidence intervals (in brackets) within each group and separately for the uteri and placentae for each cell type. Statistically significant differences between the groups within each combination of cell type and either the uteri or placentae are shown by letters, where common letters denote no statistically significant difference between the groups. Where the negative control group was excluded from the analysis, this entry was replaced by a dash (-). In cases where a robust model could not be fitted, then all entries are marked as (-) for that combination of cell type and uteri/placentae.

Tissue	Uteri				Placentae			
Group	DD	DL	LL	NC	DD	DL	LL	NC
MOMP	6.4 ^a^ (2.7–15.6)	14.2 ^a^ (6.0–34.0)	4.1 ^a^ (1.4–12.1)	-	27.1 ^a^ (18.3–40.1)	18.0 ^a^ (12.7–25.5)	27.5 ^a^ (18.6–40.6)	-
CD4 T cells	24.3 ^a^ (10.0–58.9)	32.7 ^a^ (13.5–80.0)	11.8 ^a^ (4.0–34.8)	5.4 ^a^ (1.8–15.9)	3.1 ^a^ (1.8–5.3)	0.3 ^b^ (0.1–0.9)	0.9 ^b^ (0.4–2.0)	0.4 ^b^ (0.1–1.2)
CD8 T cells	15.2 ^ab^ (6.5–35.5)	40.6 ^b^ (17.4–94.8)	9.8 ^ab^ (3.5–27.8)	4.1 ^a^ (1.5–11.7)	-	-	-	-
γδ- TCR	2.9 ^a^ (1.5–5.7)	5.0 ^a^ (2.6–9.8)	4.4 ^a^ (1.9–9.9)	1.2 ^a^ (0.5–2.8)	-	-	-	-
γδ-WC1	20.3 ^ab^ (12.1–34.2)	34.0 ^b^ (20.2–57.2)	18.7 ^ab^ (9.9–35.3)	8.8 ^a^ (4.6–16.6)	-	-	-	-
NKp46	3.3 ^ab^ (1.7–6.4)	6.7 ^b^ (3.5–12.8)	2.8 ^ab^ (1.2–6.1)	1.2 ^a^ (0.5–2.7)	-	-	-	-
TNF-α	23.2 ^b^ (7.3–73.2)	34.4 ^b^ (10.9–108.7)	20.0 ^ab^ (4.9–81.6)	2.5 ^a^ (0.6–10.0)	35.4 ^a^ (19.0–66.1)	10.9 ^b^ (6.4–18.5)	12.6 ^b^ (7.0–22.6)	6.6 ^b^ (3.6–12.2)
IL-10	8.7 ^a^ (4.6–16.1)	12.6 ^a^ (6.8–23.5)	5.1 ^a^ (2.4–10.9)	6.4 ^a^ (3.0–13.8)	2.8 ^a^ (2.1–4.0)	3.7 ^a^ (3.1–4.3)	3.2 ^a^ (2.5–4.1)	0.7 ^b^ (0.5–1.0)
FoxP3	8.5 ^a^ (4.9–15.0)	9.0 ^a^ (5.1–15.4)	10.5 ^a^ (5.4–20.4)	8.4 ^a^ (4.3–16.4)	2.3 ^a^ (1.3–4.1)	10.2 ^b^ (7.1–14.5)	5.6 ^bc^ (3.5–9.0)	4.3 ^ac^ (2.8–6.6)
IFN-γ	1.7 ^a^ (0.6–5.1)	2.2 ^a^ (0.7–6.3)	1.0 ^a^ (0.3–1.5)	-	0.4 ^ab^ (0.2–0.8)	0.6 ^b^ (0.3–1.0)	0.3 ^ab^ (0.2–0.6)	0.1 ^a^ (0.1–0.3)
IL-17A	0.5 ^a^ (0.3–0.9)	0.2 ^a^ (0.1–0.4)	0.2 ^a^ (0.1–0.4)	0.2 ^a^ (0.1–0.4)	0.3 ^a^ (0.1–0.6)	0.1 ^a^ (0.0–0.2)	0.0 ^a^ (0.0–0.1)	0.3 ^a^ (0.1–0.6)

## Data Availability

Not applicable.

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
