# Peer review of "The Immune Response in the Uteri and Placentae of Chlamydia abortus-Infected Ewes and Its Association with Pregnancy Outcomes"

_pathogens, 2023, doi:10.3390/pathogens12060846_

Round 1
Reviewer 1 Report
Caspe et al. have undertaken an important investigation into the C. abortus infection of pregnant ewes and the correlation of pregnancy success.
It is a well constructed study to investigate an understudied immune compartment of ruminants.
Major concerns:
1. Challenge controls were inoculated with C. abortus via the sub-cutaneous route. This is not the normal route of transmission and should be addressed in the disscussion. Vaginal infection would reflect the normal route of transmission.
2. Animal numbers are very low to derive statistical significance. Although I appreciate using this model would be difficult to increase numbers. This should be addressed.
Minor comments:
1. Line 164 "45 cycles" should be placed after "95oC for 15 seconds" to encompass the two previous conditions for the ease of repeatability for other researchers.
Author Response
Reviewer #1: Comments for the Authors.
Authors’ Response (AR): We thank reviewer #1 for their comments, which we have addressed below in a point-by-point response. All line numbers refer to the ‘Revised Manuscript with Track Changes’ in ‘All Markup’ view.
- Caspe et al. have undertaken an important investigation into the C. abortus infection of pregnant ewes and the correlation of pregnancy success.
It is a well constructed study to investigate an understudied immune compartment of ruminants.
AR: We thank the reviewer for their kind comments.
- Challenge controls were inoculated with C. abortus via the sub-cutaneous route. This is not the normal route of transmission and should be addressed in the discussion. Vaginal infection would reflect the normal route of transmission.
AR: The subcutaneous challenge route is well described and well documented in our many publications from over the last 30 years (e.g. see refs 3, 5, 25 and 26 in the manuscript) and as such we do not feel that this needs to be discussed or justified in this study. It is a recognized challenge animal model for C. abortus that works as effectively as the natural route of infection as we demonstrated in our 2013 study (ref https://doi.org/10.1371/journal.pone.0057950). Incidentally, the natural route of infection is actually oronasal (we do state this in the Introduction on line 42) and definitely not intra-vaginal as asserted by the reviewer. C. abortus is not a sexually-transmitted infection (e.g. see ref 2 in manuscript). However, we do recognise that we did not explicitly state that the challenge model is a well-established and described model used over the last 30 years so we have added this to lines 141-142 in the revised version.
- Animal numbers are very low to derive statistical significance. Although I appreciate using this model would be difficult to increase numbers. This should be addressed.
AR: We do highlight in several parts of the manuscript, as noted by reviewer #2, that it would be interesting to repeat the study, if possible, with larger numbers of animals because of the limitations in statistical significance. However, and again as recognised and mentioned by reviewer #2, these types of studies involving pregnant sheep models are very expensive to conduct and very complex in nature, so the use of a higher number than we have used would be considered unusual. Nonetheless, making use of existing data and samples as we have done in this study from an existing trial, is both ethically responsible and has produced some very novel data which is worth reporting. However, in view of the reviewers comment that this should be addressed we went back to our statistician who has revaluated the statistical elements of the manuscript and has made a number of significant improvements to the analysis of the datasets, resulting in improvements to the statistical comparisons. This has resulted in a number of substantial changes throughout the manuscript which as are all tracked. This also involved a replacement of Table 5 with more comprehensive data included.
- Line 164 "45 cycles" should be placed after "95oC for 15 seconds" to encompass the two previous conditions for the ease of repeatability for other researchers.
AR: The text was changed as suggested (line 187).
Reviewer 2 Report
It has been a pleasure to review this work. I congratulate the authors of this study. From my point of view, the experimental design, methodology, animal groups of study and the cell choice for the immune response study as well as the cytokines used for this study are completely accurate for the objetives.
As authors have mentioned in several parts of the manuscript, it would be interesting to repeat this type of study with a larger number of animals, but I know that this type of studies, with pregnant sheep models, are quite expensive and complex, so the use of a greater number of animals per group are unusual. Moreover, for this study authors have used the unvaccinated challenge control and negative control groups of a vaccine-challenge trial, allowing a double scientific use for these animals. I encourage the authors to continue with this line of work to clarify the complex immune response involve in EAE. Although there are many years in which this subject has been studied, it is still unclear the interactions between immune response cells and cytokines in the maternofetal interface of sheep infected with C. abortus. Results of this study are not conclusive for some of the cell populations or cytokines evaluated in this study, since no significant differences were found between the groups under study, but you have contributed with excellent figures and supplementary files that show in detail the lesions observed in the uterus and placenta of the different groups. Moreover, you have shown for the first time the role of FoxP3 and IL-10 in the pathogenesis of C. abortus in the uterus of the pregnant ewes and the irrelevant role of γδ-T cell in placentas. IHC and ISH are also highly valued. And the adequate statistical study of the results is greatly to be welcomed.
I have detected a minor writing error on line 534: the word "That" appears twice
Author Response
Reviewer #2: Comments for the Authors
Authors’ Response (AR): We thank reviewer #2 for their encouraging and kind comments. We were very pleased to hear that the reviewer valued our work.
Minor comment:
- I have detected a minor writing error on line 633: the word "That" appears twice.
AR: repetition was deleted.
Reviewer 3 Report
In this manuscript, the authors study immune cells (CD4+ T, CD8+ T, gdT and NK cells), as well as some cytokines related to the inflammatory response, present in the placenta and uterus of sheep experimentally inoculated with Chlamydia abortus. The main objective of the study was to determine the associations between the immune response at the fetal-maternal interface and the pregnancy outcomes in C. abortus infected ewes, which may contribute to understanding the pathogenicity of enzootic ovine abortion (EAE). The results are interesting since there is still much to know about the pathogenesis of C. abortus abortion and the role that the immune response could play in the induction of this abortion. However, the low number of sheep included in each experimental group makes it difficult to obtain definitive conclusions due to the great individual variability observed in the analysed parameters.
There are several aspects that are not clear or need to be improved.
In the abstract, the main aspects of the studied immune response are well summarized, such as the T helper/T reg pattern or the balance of lymphocyte subsets, but they are not mentioned in the final discussion.
The authors must revise the description of Th2 and Th1 response during pregnancy and the regulation of IFN-gamma (lines 72-77).
The description of the animals under study in 2.1 (“Ethical statement”) must be moved to 2.2. “Animals and experimental design”. Animals are selected from another unpublished experiment: 8 from the unvaccinated challenge control group and 2 negative control sheep, but no data on breed, age, or how animals are housed are given. Although challenge control animals were inoculated with the same inoculum of C. abortus strain, there were three different pregnancy outcomes. Are these different pregnancy outcomes induced experimentally or do they depend only on the individual evolution of each sheep? Could it be influenced in some way by the bacterial strain used to inoculate the animals? It is surprising that there is about a third of each pregnancy outcome, is this what usually happens in the field? In addition, the inoculation route is subcutaneous instead of the natural oronasal route. Could it have any effect on the results obtained?
Section 2.3- In sample collection, cervicovaginal mucus (CVM) swabs are missing. Also, CVM swabs must be included in the qPCR analyses (section 2.4).
Section 2.5.1. line 186. Which are the positive and negative antigen controls included in each run?
Section 2.5.3. Are primary antibodies (line 205) and primary monoclonal antibodies (line 214) the same? Are those shown in table 1? Why are primary mAbs replaced by an IgG isotype in NC sections?
Section 2.5.4- Figure S1 corresponds to an IHC in Caprine fetal small intestine infected with C. pecorum, not to an IHC in C. abortus positive and negative control placenta samples. Review and modify lines 230-233.
Figures S2 and S3 must be included in the manuscript in addition to figures 2 and 4.
Table 1. The tittle should include not only the labelling of immune cells but also the detection of Chlamydial MOMP antigen.
Section 2.5.5 and Table 2. Check the name of the probes.
Table 3- Which is the difference between NF and O in macroscopic lesions (%)? What is the meaning of NF in qPCR? The results of mZN in CVM swabs are missing (see also line 338)
Analysis of NC placenta samples by qPCR results in the detection of low levels of chlamydial DNA, not zero (lines 341-343). How can it be explained? And how is it related to negative IHC results in tissue sections? (Lines 362-363 and figure 7).
Lines 354-355. CVM samples was also taken for NC sheep.
Results from figure 5 and table 4 are a bit confusing (“in the placentae, infiltration of CD4+T cells, CD8+T cells, γδ-T cells and NK cells was occasionally present in small numbers or in single cell” (figure 5), but in table 4 all IHC results in DD, DL, LL (and NC) are zero except in gd-WC-1 DL. The interpretation of score of IHC (Q1, M, Q3) should be included in material and methods (2.5.6). IHC markers should have the same name in the figures and in Table 4 ( gdT-TCR instead of gdT-86D; gdT-WP-1 instead of gdT-CC15).
ISH of IFN-g mRNA: it is difficult to see the higher presence in the placenta in DL because it appears similar to NC (figure 6). Results from IL-17 mRNA are not shown. Figure S4 allows a better comparison between groups, it should be included in the manuscript.
Figure 7. What are Ut.or.PI (bottom right of the figure)? There are too many outlier data, is it difficult to interpret the results. Can it be explained by the low number of animals in each group? In fact, there are very few significant differences in uteri and placenta (sections 3.8 and 3.9).
Lines 473-575. Results of qPCR in placenta LL are confusing, are they the lowest or the highest?
The conclusions are somewhat ambiguous regarding the presence and role of immune cells and cytokines in the uterus and placenta in C. abortus infected sheep. The authors adequately explain the need to conduct research with a larger number of animals to validate the results of this study.
Author Response
Reviewer #3: Comments for the Authors
Our responses to the Reviewer’s comments are detailed below. All line numbers refer to the ‘Revised Manuscript with Track Changes’ in ‘All Markup’ view.
- In this manuscript, the authors study immune cells (CD4+ T, CD8+ T, gdT and NK cells), as well as some cytokines related to the inflammatory response, present in the placenta and uterus of sheep experimentally inoculated with Chlamydia abortus. The main objective of the study was to determine the associations between the immune response at the fetal-maternal interface and the pregnancy outcomes in C. abortus infected ewes, which may contribute to understanding the pathogenicity of enzootic ovine abortion (EAE). The results are interesting since there is still much to know about the pathogenesis of C. abortus abortion and the role that the immune response could play in the induction of this abortion.
AR: We thank the reviewer for their kind comments.
- In the abstract, the main aspects of the studied immune response are well summarized, such as the T helper/T reg pattern or the balance of lymphocyte subsets, but they are not mentioned in the final discussion.
AR: These observations were mentioned in the Discussion on lines 771-791.
- The description of the animals under study in 2.1 (“Ethical statement”) must be moved to 2.2. “Animals and experimental design”. Animals are selected from another unpublished experiment: 8 from the unvaccinated challenge control group and 2 negative control sheep, but no data on breed, age, or how animals are housed are given. Although challenge control animals were inoculated with the same inoculum of C. abortus strain, there were three different pregnancy outcomes. Are these different pregnancy outcomes induced experimentally or do they depend only on the individual evolution of each sheep? Could it be influenced in some way by the bacterial strain used to inoculate the animals? It is surprising that there is about a third of each pregnancy outcome, is this what usually happens in the field? In addition, the inoculation route is subcutaneous instead of the natural oronasal route. Could it have any effect on the results obtained?
AR: Re-reading the ethical statement we can understand the point the reviewer is making here. We slightly reorganised the text to make it clearer that the animals originated from another study and have moved the section as suggested by the reviewer to the “Animals and experimental design” section (see lines 128-145). Data on breed, age and serological testing was included (lines 146-148). The animals are challenged and thus the infection is experimentally induced. We have added in the detail of this challenge (lines 139-142). The model we use is a well-characterised and established model that our group has used for over 30 years and produces disease indistinguishable from an experimental oro-nasal infection that mimics a natural infection. We have added this point (lines 141-142).
- Section 2.3- In sample collection, cervicovaginal mucus (CVM) swabs are missing. Also, CVM swabs must be included in the qPCR analyses (section 2.4).
AR: Our apology for this error, we have now included CVM in sections 2.2 and 2.3 (previously 2.3 and 2.4) (lines 173-174; 177-178).
- Section 2.5.1. line 186. Which are the positive and negative antigen controls included in each run?
AR:. This information was included in original section 2.5.3 (now 2.4.3 Immunolabelling of tissue sections), which we have now moved to this section (now 2.4.1).
- Section 2.5.3. Are primary antibodies (line 205) and primary monoclonal antibodies (line 214) the same? Are those shown in table 1? Why are primary mAbs replaced by an IgG isotype in NC sections?
AR: Yes these are the same and yes these are in Table 1 as stated, clarified in text on line 230. The antibodies used for the negative control slides are non-specific (ie not specific for Chlamydia) mAbs of the same isotype, according to standard practice.
- Figures S2 and S3 must be included in the manuscript in addition to figures 2 and 4.
AR: The information in figures S2 and S3 is included in Figure 7. Figures S2 and S3 show the individual data points, while Figure 7 is a summary for all markers in both placentae and uteri making it easier to visualise the data for all markers in the two tissues in the same figure rather than individual ones. Hence, for completeness we placed the individual figures as supplementary files. There is no point in duplicating this information in the main paper.
- Table 1. The tittle should include not only the labelling of immune cells but also the detection of Chlamydial MOMP antigen.
AR: The title for Table 1 has been amended.
- Section 2.5.5 and Table 2. Check the name of the probes.
AR: We have checked the website and can confirm that we did indeed get the names slightly wrong for which we apologise. These have been corrected. The probes used are those listed under Cat No. 1048671-C1 (RNAscopeTM Probe - Oa-IL17A-C1) and Cat No. 495621 (RNAscopeTM Probe - Oa-IFNG-XCh).
- Table 3- Which is the difference between NF and O in macroscopic lesions (%)? What is the meaning of NF in qPCR? The results of mZN in CVM swabs are missing (see also line 338)
AR: The “NF” means that the placenta was not found and “0” means that there were no visible lesions observed. We do not perform mZN smears on CVM swabs, the error has been corrected in the manuscript (line 391). NF for qPCR has been changed to NA, being not applicable as no placentas were available.
- Analysis of NC placenta samples by qPCR results in the detection of low levels of chlamydial DNA, not zero (lines 341-343). How can it be explained? And how is it related to negative IHC results in tissue sections? (Lines 362-363 and figure 7).
AR: The non-zero result in the qPCR is related to the extremely high sensitivity of the technique and low levels of environmental contamination, leading to the apparently low levels detected, which are below our limit of detection of 100 genome copies per reaction (publication in preparation).
- Lines 354-355. CVM samples was also taken for NC sheep.
AR: The sentence was modified (lines 411-412)
- Results from figure 5 and table 4 are a bit confusing (“in the placentae, infiltration of CD4+T cells, CD8+T cells, γδ-T cells and NK cells was occasionally present in small numbers or in single cell” (figure 5), but in table 4 all IHC results in DD, DL, LL (and NC) are zero except in gd-WC-1 DL. The interpretation of score of IHC (Q1, M, Q3) should be included in material and methods (2.5.6). IHC markers should have the same name in the figures and in Table 4 ( gdT-TCR instead of gdT-86D; gdT-WP-1 instead of gdT-CC15).
AR: We apologise if we have not been clear on the definition of quartiles and medians in the context of this study. We have added an explanation in the materials and methods under Statistical Analysis as requested (see section 2.5). We have amended the inconsistency of naming of IHC markers in summary Table 4 and the figures to ensure consistency and so that they link directly to Table 1.
- ISH of IFN-g mRNA: it is difficult to see the higher presence in the placenta in DL because it appears similar to NC (figure 6). Results from IL-17 mRNA are not shown. Figure S4 allows a better comparison between groups, it should be included in the manuscript.
AR: We recognise that the photos selected do not wholly match the description, therefore we have modified figures 6 to reflect this. As we mentioned under point 7, the comparison is summarised in figure 7 and hence including figure S4 as a supplementary file to avoid repetition. Therefore our preference is to leave it as it is.
- Figure 7. What are Ut.or.PI (bottom right of the figure)? There are too many outlier data, is it difficult to interpret the results. Can it be explained by the low number of animals in each group? In fact, there are very few significant differences in uteri and placenta (sections 3.8 and 3.9).
AR: The key for figure 7 has been updated. As stated by the reviewer under point 17 below, we have clearly recognised the limitations of this study, which could be addressed further using larger animal numbers and the use of spatial transcriptomics. This was discussed in the conclusion section (lines 839-846). Nevertheless, the present study did produce novel data that suggests there could be differences in immune composition relating to pregnancy outcome following experimental C. abortus infection. However, in view of the reviewers comments regarding outliers and interpretation of results we went back to our statistician who has revaluated the statistical elements of the manuscript and has made a number of significant improvements to the analysis of the datasets, resulting in improvements to the statistical comparisons, particularly between placentae and uteri. This has resulted in a number of substantial changes throughout the manuscript which as are all tracked. This also involved a replacement of Table 5 with more comprehensive data included. We hope the reviewer agrees that this has improved the manuscript.
- Lines 473-575. Results of qPCR in placenta LL are confusing, are they the lowest or the highest?
AR: We assume the reviewer meant lines 573-575. The text has been corrected (lines 674-679).
- The conclusions are somewhat ambiguous regarding the presence and role of immune cells and cytokines in the uterus and placenta in C. abortus infected sheep. The authors adequately explain the need to conduct research with a larger number of animals to validate the results of this study.
AR: As mentioned under point 15, we acknowledge that the reviewer agrees with our conclusions of the limitations of this initial investigation.
Round 2
Reviewer 3 Report
I thank the authors for the revised version. They have corrected the mistakes and adequately answered my previous questions. I agree not to include the figures from the supplementary material in the manuscript. The manuscript has improved remarkably. However, I still have two observations.
Figure S1 is not included in the manuscript, only in the supplementary material. It is not clear why it is necessary to provide this figure, as it shows immunolabelled images of the small intestine of fetal goats infected with C. pecorum. It should be better explained in the text of the material and methods section.
I have not been able to find the observations on the balance of lymphocyte subsets or the T helper/T reg pattern that the authors have included in the Discussion at lines 771-791.
Author Response
Response to Reviewers
Our responses to the Reviewer’s comments are detailed below.
Reviewer #3 (round 2): Comments for the Authors.
- I thank the authors for the revised version. They have corrected the mistakes and adequately answered my previous questions. I agree not to include the figures from the supplementary material in the manuscript. The manuscript has improved remarkably.
AR: We thank the reviewer for their kind comments.
- Figure S1 is not included in the manuscript, only in the supplementary material. It is not clear why it is necessary to provide this figure, as it shows immunolabelled images of the small intestine of fetal goats infected with C. pecorum. It should be better explained in the text of the material and methods section.
AR: explanation was added in the Material and Methods section (lines 238-239).
- I have not been able to find the observations on the balance of lymphocyte subsets or the T helper/T reg pattern that the authors have included in the Discussion at lines 771-791.
AR: Thank you for the observation. Although it was not discussed directly with the balance of T helper/Treg, in the discussion (lines 741-749), the findings related to Treg were discussed as their internal cellular marker (FoxP3) and their main cytokine (IL-10), and an additional reference for the CD4 marker is included in lines 745 and 746. This combination of single marker staining has been used for the first time here for experimental EAE. The overlap of CD4 for Treg and T helper cells and IL-10 for Treg and antigen-presenting cells makes a further assignment of cell phenotype balance problematic and liable for incorrect overinterpretation without further analysis, such as dual antibody labeling or spatial transcriptomics. Such future work and directions have been respectfully suggested in lines 798-805.
